

# A configurable simulator for low-resolution infrared thermal sensors: accuracy assessment and practical applications in indoor environments

Sara Comai[1], Andrea Masciadri[2], Andrea Locati[2], Alessandro Campi[1] and Fabio Salice[1]

[1] Department of Electronics Information and Bioengineering, Politecnico di Milano, Milano, Italy
[2] Politecnico di Milano, Milan, Italy

## ABSTRACT

Thermal sensors are increasingly used in various applications such as environmental monitoring, smart homes, and surveillance. These sensors detect infrared radiation to monitor human presence and movement, enabling sophisticated sensing capabilities. However, determining the number of sensors and optimizing their placement through real-world experimentation is often impractical due to cost and logistical constraints. This article introduces *ThermalSim*, a highly configurable simulator for low-resolution thermal sensors, designed to address these issues and enable pre-testing of sensor setups in virtual environments. The simulator allows for detailed customization of environmental parameters, object properties, and sensor characteristics and supports modelling a dynamic agent with configurable trajectories and speeds. Experimental validation against real-world data demonstrates the simulator's high accuracy in replicating static and dynamic scenarios. Metrics such as correlation, entropy and mutual information, and similarity of temperature images have been used to evaluate the simulator's output. Some case studies show the tool's flexibility, showcasing its practical applications across various scenarios.

# INTRODUCTION

Sensing technologies are widely used to monitor human behaviour and environmental conditions. Their applications include smart homes, health monitoring, security, and energy efficiency. These technologies can be classified into intrusive and non-intrusive methods, with non-intrusive sensors—such as temperature, power, and position sensors—gaining attention due to their ability to collect meaningful data without requiring direct user interaction (*Fu et al., 2020*; *Bian et al., 2022*). While wearable devices offer precise health tracking capabilities, they come with challenges such as user compliance and frequent maintenance, motivating the exploration of alternative sensing approaches.

Corresponding author
Sara Comai, sara.comai@polimi.it

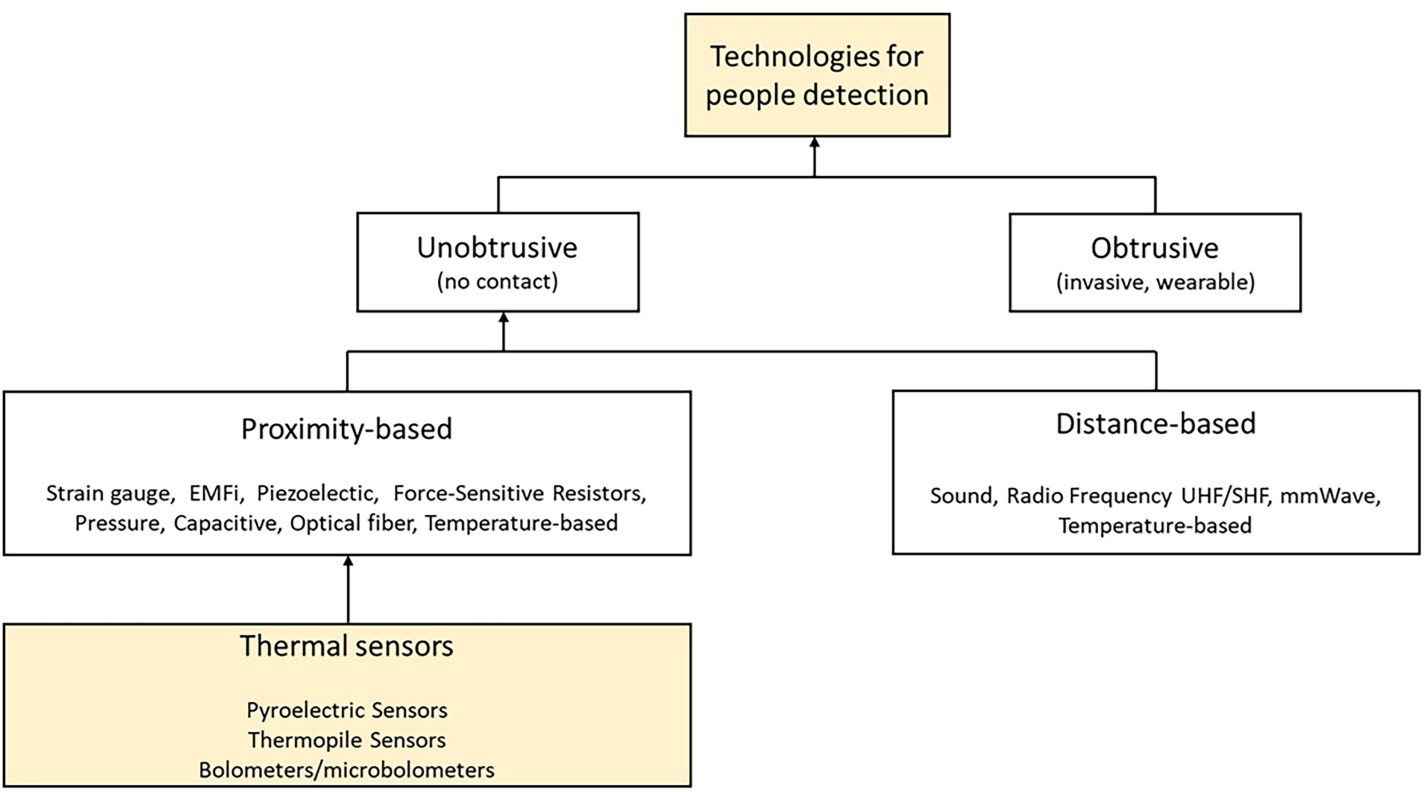

**Figure 1** **Taxonomy of the sensing techniques for people detection with a focus on thermal sensors and their characteristics.**

Unobtrusive monitoring can be achieved using a range of proximity-based and distance-based sensors (Fig. 1). Proximity-based sensors are effective for detecting nearby activity, while distance-based sensors provide broader coverage, making them suitable for various environments. Among these, temperature-based sensors stand out due to their versatility, affordability, and privacy-preserving characteristics (*Nwakanma et al., 2023*).

Thermal sensors, in particular, have emerged as a promising solution to monitor human presence and movement based on infrared radiation (*Comai et al., 2025*). Their advantages—including non-intrusiveness, low sensitivity to lighting conditions, and compact design—make them well-suited for applications that prioritize privacy, cost-effectiveness, and reliability (*Nwakanma et al., 2023*).

Thermal sensors come in different types, each with distinct sensing mechanisms (*Moisello, Malcovati & Bonizzoni, 2021*), resolutions and prices. Pyroelectric sensors (PIR—Passive Infrared) detect temperature changes in radiant energy and are widely used for motion detection. Thermopile-based temperature sensors (*e.g.*, Panasonic Grid-EYE) continuously monitor the temperature of radiant energy in their field of view, offering more detailed thermal data for the detection of both motion and stationary objects (*Chuah & Teoh, 2020*). For this reason, they are being increasingly used in advanced applications such as smart buildings, presence detection, and people counting. Finally, with higher resolutions and costs, (micro)bolometers detect infrared radiation by converting temperature-induced resistance changes in an embedded resistor into measurable voltage variations. They are typically used in military/marine systems, automotive, thermography and surveillance (*Moisello, Malcovati & Bonizzoni, 2021*).

Despite their advantages, deploying thermal sensors—in particular thermopile-based temperature sensors—in real-world scenarios presents significant challenges. Identifying the optimal placement and proper sizing of a sensor network, along with evaluating its efficiency and coverage, is a time-consuming and costly process (*Kim & Jo, 2024*). Factors such as physical obstructions, environmental conditions, and varying user behaviours can impact sensor performance and data accuracy, making real-world experimentation costly and logistically difficult.

Simulations offer a controlled, replicable environment where various parameters—such as room layout, sensor placement, and human activity—can be defined with precision. This flexibility allows testing sensors' performance under different conditions to maximize coverage, accuracy, and efficiency without setting up physical experiments for every scenario (*Golestan et al., 2019*; *Almutairi, Bergami & Morgan, 2024*; *Kim & Jo, 2024*).

However, existing smart home simulators typically do not support thermal sensor data generation, limiting the ability to simulate and optimize sensor networks that use these complex data types. Furthermore, current simulators rarely tackle the problem of sensor placement optimization to ensure comprehensive coverage and minimal blind spots.

To overcome these challenges, we introduce *ThermalSim*, a configurable simulator specifically designed for *low-resolution* thermal sensors. Developed in MATLAB (The MathWorks, Natick, MA, USA) the simulator can reproduce the behaviour of low-resolution thermal sensors. The simulator is fully configurable. Users can set parameters such as resolution, field of view (FOV), noise level, operating distance, and detectable temperature range of each sensor. The virtual environment is then represented visually and the temperature images of each sensor are automatically generated, as depicted in Fig. 2.

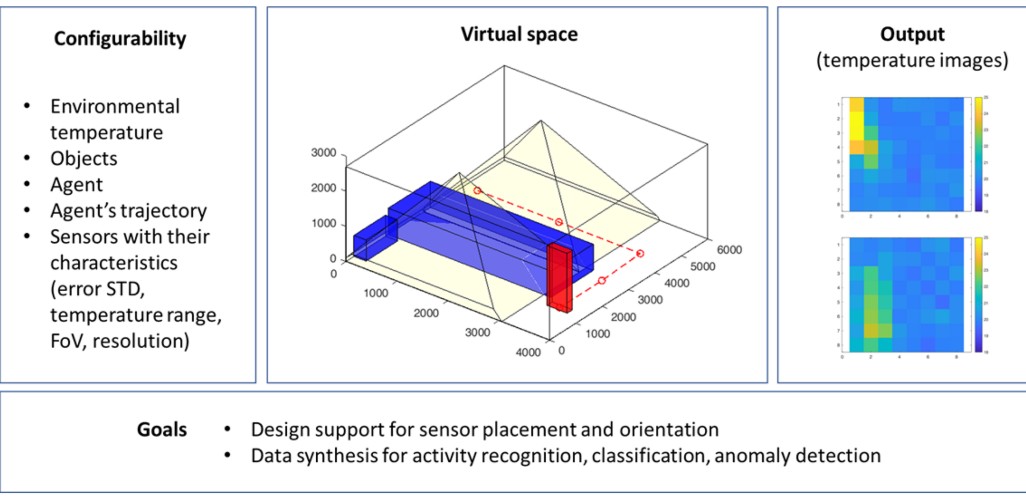

**Figure 2** *ThermalSim*: configurable components, virtual environment, sample output, and main goals.

The main objectives of this work are twofold:

(i) To support the design and pre-validation of sensor networks by simulating various sensor placements and orientations in indoor environments;

(ii) To provide synthetic data streams for testing algorithms in activity recognition, classification, or anomaly detection.

For the experimental evaluation, we selected the Grid-EYE ($8 \times 8$) sensor to show the accuracy of the simulator in replicating real-world sensor performance. This sensor is widely used to detect motion, presence, and relevant spatial patterns, due to its cost-effectiveness, computational simplicity (ideal for embedded or real-time systems), and sufficient spatial granularity for the above tasks. We also present an experiment with the FLIR Lepton $80 \times 60$ thermal camera, to show how the simulator scales to higher resolutions. Many low-resolution sensors fall within the range of these two sensors.

To demonstrate its practical utility, we present two case studies: the first illustrates a study on how multiple sensor arrays can enhance spatial resolution through a super-resolution effect; the second explores optimal sensor placement strategies for reliable presence detection in specific monitoring tasks, such as bed monitoring. By using multiple cameras, we demonstrate the simulator's scalability across different sensor configurations.

The article is organized as follows: 'Related Work' presents related work. 'Overview of the Simulator' provides an overview of the simulator and explains how it replicates real-world environments and sensor behaviour. 'Metrics for the Evaluation of the Simulator' introduces key metrics such as correlation, mutual information, and image similarity to evaluate the simulator's accuracy of the generated data compared to real-world data. 'Experiments and Results' presents experimental results. 'Scenario

Analysis' illustrates two scenarios where the simulation can be applied. Finally, 'Areas for Further Development' discusses areas for further development of the simulator, while 'Conclusions and Future Works' concludes the article.

# RELATED WORK

**Thermal sensors for human detection and classification.** Thermal sensors have been applied in various contexts, from activity recognition to motion tracking (*Nwakanma et al., 2023*). A number of works demonstrate the feasibility of using low-resolution thermal arrays for human detection and posture classification. *Yin et al. (2021)* used a low-resolution infrared array (Grid-EYE $8 \times 8$) and a deep-learning framework based on long short-term memory (LSTM) for human activity recognition such as lying, sitting, walking, and standing. *Hand et al. (2022)* achieved an accuracy of 99.7% in detecting bed occupancy using a low-resolution ceiling-installed thermal sensor, showing the effectiveness of thermal data for static presence detection. *Puurunen et al. (2021)* used an Omron D6T-44L06 thermal sensor to detect human presence with an artificial neural network (ANN), achieving an accuracy of 99.6%. *Singh & Aksanli (2019)* used two vertically stacked MLX90621 sensors to detect and count individuals (up to three) and recognize static postures such as sitting and standing, achieving 97.5% accuracy for activity classification and 100% for presence detection.

**Advancements in thermal sensing algorithms and optimization methods.** Other studies have focused on algorithmic enhancements to improve sensing robustness. *Trofimova et al. (2017)* enhanced detection algorithms by addressing environmental temperature variations using adaptive background estimation and noise removal techniques based on Kalman filters. The results demonstrate improved human detection accuracy (from 70% to 97.0%) over baseline techniques (*Mashiyama, Hong & Ohtsuki, 2015*), particularly in noisy environments. Also *Yin et al. (2021)* incorporated temperature variation filtering as a preprocessing step to improve recognition accuracy in classifying lying, sitting, walking, and standing postures. *Shetty et al. (2017)* proposed a comprehensive processing pipeline to detect and track humans using the Grid-EYE $8 \times 8$ thermopile array sensor. Their method starts with bicubic interpolation to enhance the spatial resolution of the raw $8 \times 8$ thermal frames, followed by background subtraction to isolate foreground regions. These are further refined through Gaussian filtering and an iterative threshold selection algorithm to produce a binary map of human presence. For motion tracking, centroid estimation combined with a Kalman filter was used to predict human trajectories. Their system successfully tracks both static and moving individuals, compensating for the sensor's low resolution through a layered processing approach.

The previous works demonstrate that, with appropriate processing, even low-resolution thermal arrays can be highly effective in detecting and tracking people. However, most focus on signal-level improvements, assuming fixed sensor configurations.

**Gaps in deployment and the role of simulation.** While these studies demonstrate the effectiveness of thermal sensors across various applications, a critical deployment-related question remains largely unanswered: how should thermal sensors be placed and

configured in complex indoor environments to ensure optimal performance? The studies reviewed generally assume a single sensor or do not discuss layout optimization, coverage trade-offs, or sensitivity to spatial arrangement. In real-world applications, furniture, room geometry, and multiple occupants can significantly affect sensor effectiveness.

Our work addresses this gap by proposing a simulation-based framework to evaluate and optimize thermal sensor configurations.

**Limitations of existing smart home simulators.** Numerous simulators have been developed to generate datasets for supporting smart home design, as highlighted in the review in *Alshammari et al. (2017)*. These simulators typically focus on classifying activities of daily living and detecting anomalous behaviours (*Alshammari et al., 2018*; *Bouchard et al., 2010*; *Comai et al., 2023*). They produce simulated sensor data that mimic the output of various sensors deployed in a smart home environment. Commonly included sensors are motion detectors such as passive infrared (PIR), door and contact sensors, pressure sensors (*e.g.*, those activated by beds or couches), and appliance switches. Thermal sensors generate more complex output data and, for this reason, are not integrated into these types of simulators. Furthermore, such simulators typically do not address the problem of determining the optimal placement of sensors to ensure adequate coverage and minimize blind spots.

# OVERVIEW OF THE SIMULATOR

*ThermalSim* supports the design and development of monitoring systems that use low-resolution thermal sensors. Its purpose is to study how various factors affect the sensor's detection capabilities, including sensor positioning (viewing angle, height, *etc.*), the temperature of both objects and the surrounding environment, as well as the movement of an agent or moving objects within the monitored space.

Developed in the Matlab environment, it allows users to create virtual spaces with detailed environmental settings, including customizable room dimensions, objects with defined thermal properties, and human agents with adjustable parameters. The simulator replicates the real-world behaviours of thermal sensors, enabling the configuration of sensor properties such as resolution, field of view, and noise characteristics. Figure 3 shows an example of a room with the main elements: two blue objects representing a table and a fan coil, a red element representing an individual, and two yellow cones representing the field of view of the sensor.

**Environment configuration.** The space where the simulation occurs is a three-dimensional space that conceptually represents a room in which agents or objects can be positioned. This environment is characterized by properties such as room dimensions (width, length, and height) and a uniform average temperature. The coordinate system is set with the origin at the bottom-left corner of the space, providing a reference for positioning agents and objects. The simulator assumes a constant temperature throughout the space and does not simulate temperature gradients or variations within different areas of the room.

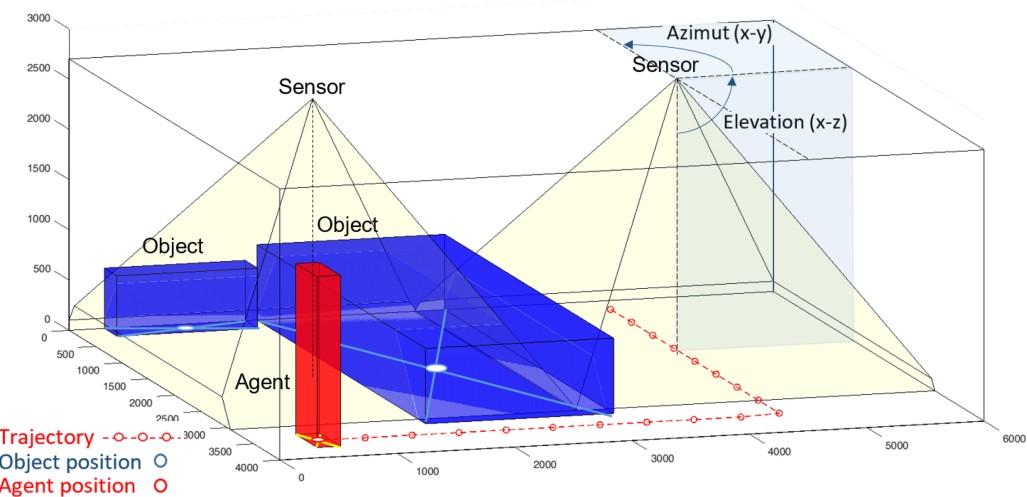

**Figure 3** Example of a 3D room (4,000 × 6,000 × 2,700 (mm³)) with two (blue) objects representing a large table and a fan-coil, a (red) agent representing a standing person and two sensors ceiling-mounted sensors positioned at a height of 2,700 mm with a 45° elevation and a −90° azimuth with field of views represented by the yellow pyramids. The red dotted line indicates the agent's trajectory during the simulation.

**Sensors configuration and placement**. The simulator enables the placement and configuration of sensors within the environment, allowing users to model various sensor types. Each sensor is defined by its position, field of view (FOV), resolution, operating distance, and noise characteristics. The simulator accurately simulates sensor behaviour, reflecting real-world conditions by capturing the average temperature within each sensor's FOV. A noise model is also incorporated to account for sensor inaccuracies, tailored to the specific characteristics of each sensor.

**Object placement**. Users can introduce one or more objects into the environment, with each object characterized by its position, shape, temperature, and dimensions. The simulator supports a variety of shapes, including cuboids, pyramids, and custom-defined shapes. Objects must be positioned within the defined boundaries of the space; any object extending beyond these limits will be automatically removed. By default, the objects are displayed in blue, though the colour can be adjusted within the simulator.

**Agent configuration, placement and trajectory definition.** The simulator supports the inclusion of a single mobile human agent, characterized by attributes such as name, body dimensions, average temperature, and movement speed. The agent's motion follows a trajectory specified in a .csv file, allowing the simulator to interpolate between points and create a smooth, continuous movement path. Multiple static human agents can be represented as objects within the environment, each assigned a specific temperature to simulate their thermal presence. By default, the agent is displayed in red, though the colour can be modified in the simulator.

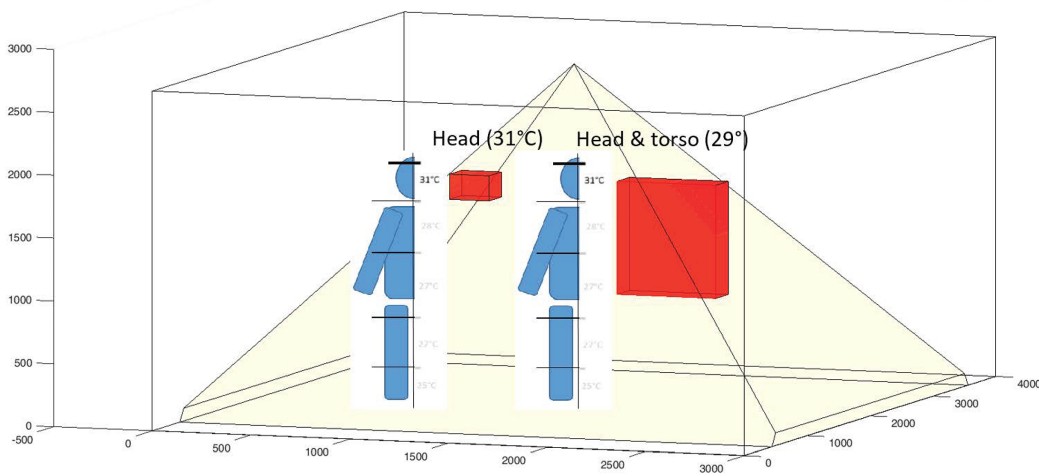

**Figure 4 Example of two body models representing only the exposed parts of individuals detectable by the sensor.** Human figures are included for visual clarity.

Figure 4 illustrates a scenario with two agents. Since the sensor cannot detect body parts covered by insulating clothing, only the exposed areas of the agents are represented as corresponding parallelepipeds (*e.g.*, the face and parts covered by a sweater, but not the areas covered by boots or insulating clothing). The exposed parts are assigned a temperature that reflects the average surface temperature.

The output of the simulation consists of two components: a visual representation of the environment, as shown in Figs. 3 and 4, and a sequence of views from the sensors, which are presented later in the article.

## Grid-EYE sensor in the case examples

Without loss of generality of the approach, almost all the examples throughout the article and the validation processes consider a Panasonic Grid-EYE 8 × 8 sensor, equipped with 64 thermopile elements. Like other sensors belonging to this family, it detects the absolute temperature at each thermopile, allowing the generation of thermal images based on actual temperature readings. The sensor features a built-in silicon lens with a 60° viewing angle (FOV—field of view), making it suitable for monitoring a medium-sized room. For example, when positioned at a height of 2.7 m (the minimum height of a room in Italy), the covered area is 3 × 3 square meters, with each pixel representing a 40 × 40 cm$^2$ section. The Grid-EYE sensor measures temperatures ranging from 0 °C to 80 °C with an accuracy of ±2.5 °C. It can detect human bodies at distances of up to 7 m (its operating distance). Data are transmitted *via* an I$^2$C interface at a rate of 1 or 10 frames per second, with an interrupt signal available for timely event responses.

## Sensor model implementation: from the model to the images

The sensor can be configured based on the following parameters: orientation, resolution, and field of view (FOV). Its geometric model is a three-dimensional quadrangular pyramid

with its vertex positioned at the sensor location, its main axis determined by two angles—elevation and azimuth—and a base whose dimensions depend on the FoV's angular width. This pyramid represents the volume of the space observable by the sensor. A projection (or vision) plane is placed in front of the vertex, at a distance proportional to the resolution and FOV's angular width. The FOV is divided into one-degree units, both horizontally and vertically, forming pyramid trunks. Each trunk is a truncated portion of pyramids with a vertex angle of 1° in the two directions. Each trunk is further subdivided into smaller parts, according to the parameter `pixels_per_degree`, which determines the granularity of the division and the resolution of the projection plane. It affects the precision of the measurements and the computational performance.

For example, with a FOV of 60° in each direction and 5 `pixels_per_degree` (the default value), the total number of pyramid trunks is $60 \cdot 5 \times 60 \cdot 5 = 300 \times 300$. This implies a resolution of $300 \times 300$ points. The distance $h$ of the projection plane from the pyramid's vertex is calculated using the formula:

$$h = \frac{\text{FOV}}{2} \cdot \text{pixels\_per\_degree} \cdot \cot\left(\frac{\text{FOV}}{2}\right)$$

For a FOV $= 60°$ and 5 pixels per degree, the distance of the vision plan is $h \approx 77.94\,\text{mm}$.

The pyramid trunks intersect the objects and the avatar present in the environment, modelled as polygons. Portions of objects and the avatar that are occluded by other objects are removed. The visible polygons are then projected onto the sensor's vision plane, with each point associated with the distance and temperature of the object. For each point in space, the algorithm:

1. Computes the intersection of the line connecting the point and the sensor with the sensor's view plane.
2. Rotates the projected point to align it with the sensor's orientation.
3. Translates the point into the sensor plane's coordinates, adjusting for FOV and resolution.
4. Generates the 2D coordinates $(x, y)$ of the projected point on the sensor plane.

### Perceived temperature calculation

The distance of the polygon point from the sensor's projection plane, along with the corresponding temperature, is used to compute the *perceived temperature*. The model is derived from the Stefan-Boltzmann (SB) law, which describes the heat transfer by radiation for a body immersed in air at temperature $T_{\text{ambient}}$. According to the SB law, the perceived temperature at a distance $d$ depends on the distribution of radiated energy. Radiated energy decreases with the square of the distance, while the energy emitted by a blackbody (or nearly blackbody) is proportional to the fourth power of its absolute temperature. The following formula is a good approximation of the perceived temperature, as confirmed by experimental results. It is important to note that, at large distances, the ambient temperature dominates the perceived temperature, while the object's temperature becomes more influential at close distances.

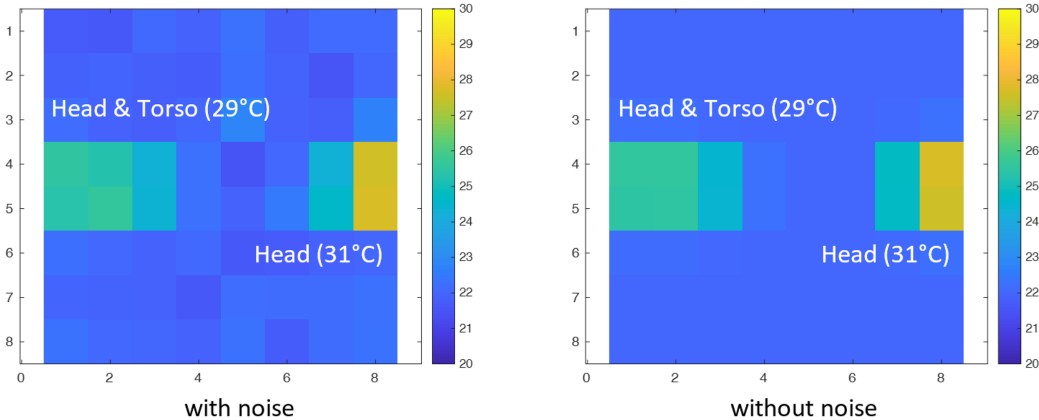

**Figure 5** Simulated sensor images of the model depicted in **Fig. 4**, with and without noise (Grid-EYE 8 × 8, FOV 60°).

$$T_{\text{perceived}} = \left( T_{\text{ambient}}^4 + \frac{\varepsilon \cdot (T_{\text{object}}^4 - T_{\text{ambient}}^4)}{\text{distance}^2} \right)^{\frac{1}{4}}.$$

In this equation, the parameter $\varepsilon$ represents the emissivity of the materials and is typically assigned a value of 0.8, which is suitable for opaque or dark surfaces such as fabrics. To avoid unrealistic values, the model enforces a minimum distance of 1 cm from the sensor.

For distances shorter than 1 cm, a linear model is used, where the perceived temperature decreases linearly with distance:

$$T_{\text{perceived}} = T_{\text{ambient}} + \left( 1 - \frac{\text{distance}}{\text{operational\_distance}} \right) \cdot \left( T_{\text{object}} - T_{\text{ambient}} \right)$$

where operational_distance is the maximum operational distance of the sensor (default: 5 m).

### Interpolation

The generated 300 × 300 matrix is resized to the final resolution of the sensor (*e.g.*, 8 × 8 for the Grid-EYE sensor) using bilinear interpolation.

### Noise

To consider the imperfections and limitations of the sensor, the accuracy of the temperature (the characteristic sensor error) is derived from the sensor's datasheet and modelled as a Gaussian noise with a zero mean and a standard deviation, which is added to the temperature value. The standard deviation is a parameter: for example, 0.2 is used for the Grid-EYE sensor.

Figure 5 shows the images generated by the simulator for the model represented in Fig. 4, as seen from the sensor: the image on the left-hand side does not include the noise model; the image on the right-hand side includes the Gaussian noise with standard deviation 0.2. Notice that the object temperatures and the sensed temperatures differ due

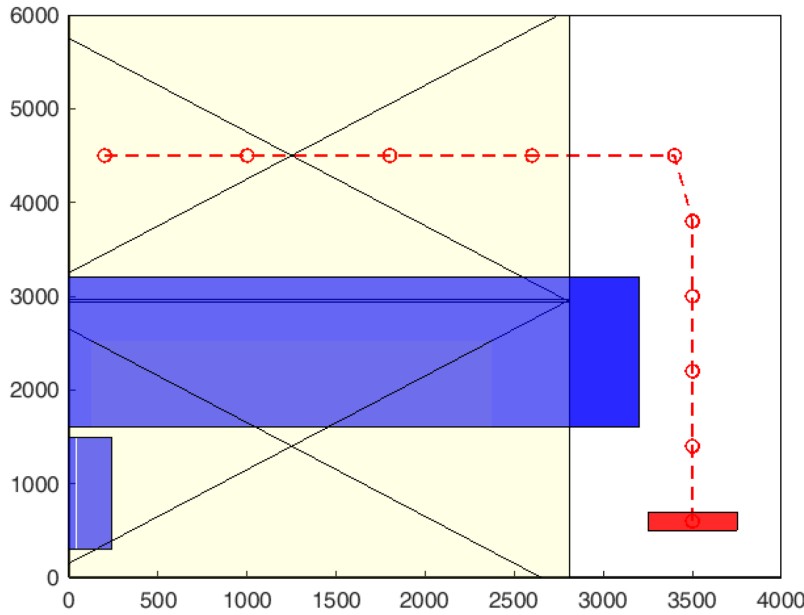

**Figure 6 Simulation with a 1 Hz sampling rate and an avatar speed of 800 mm/s.** The trajectory is represented by discrete points.

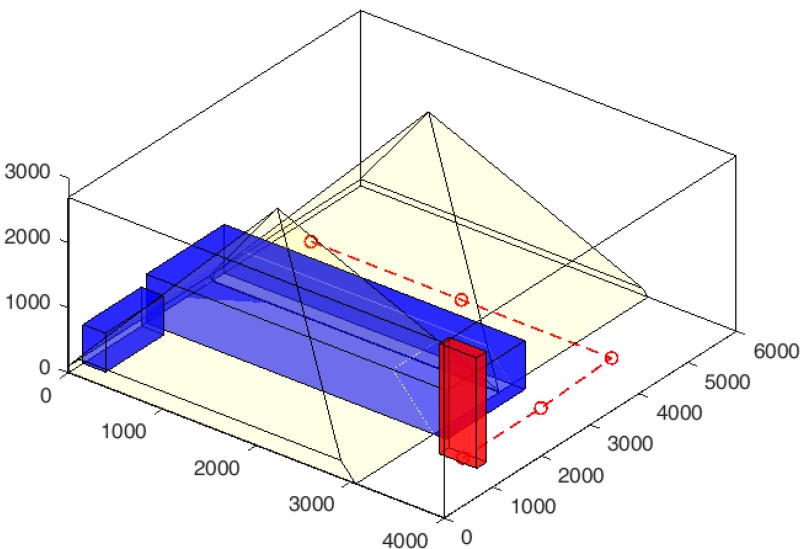

**Figure 7 Simulation with a 1 Hz sampling rate and an avatar speed of 1,600 mm/s.** The trajectory points are more spaced out as the avatar moves faster, covering a greater distance between samples.

to distance and surface properties. For example, a head at 31 °C is detected as an object with a temperature between 28 °C and 29 °C.

### Sensor sampling frequency

For moving objects, like in the simulation in Fig. 3, the sensor's sampling frequency is specified.

It determines the points along the avatar's trajectory where thermal images are captured. The avatar's position is calculated based on both its speed (a parameter of the agent) and the sampling frequency. For example, if the sampling frequency is 1 Hz and the avatar's speed is 0.1 m/s, the simulator captures an image every 0.1 m along the avatar's path.

Figures 6 and 7 show the images generated in two simulations, using a sampling frequency of 1 Hz with avatar speeds of 800 and 1,600 mm/s, respectively. The environment contains two Grid-EYE sensors, two objects and an agent as in Fig. 3. The circles drawn by the simulator along the avatar's trajectory represent the points where the sensor captures images. With the higher speed, the points are more sparse.

## METRICS FOR THE EVALUATION OF THE SIMULATOR

The performance of the simulator was evaluated by comparing its output with real-world data collected from actual sensors. The primary objective was to validate the simulator's accuracy in replicating the behaviour of thermal sensors under both static and dynamic conditions. This evaluation involved recreating a real-world environment within the simulator, generating simulated sensor data, and comparing it with data obtained from physical sensors. To quantify the accuracy of the simulation results, metrics such as correlation, mutual information, and image similarity were employed. The following sections provide a detailed description of the real-world environment setup, the simulation process, and the comparison methodology.

### Setup of the real world environment

Figure 8 shows an overview of the system architecture used for the validation, while Fig. 9 depicts the component diagram. One or more Grid-EYE sensors are connected to a Raspberry Pi *via* the I$^2$C interface. The data collected by the Raspberry Pi are transmitted *via* Wi-Fi to a Web server running on a PC. The PC serves the dual purpose of storing the data and providing a user interface for viewing it. The server-side framework used for developing the Web application is Flask. The detection rate of the Grid-EYE is 1 to 10 frames per second (each frame being an $8 \times 8$ matrix of 14-bit values for each thermopile). Every 0.5 s, each Raspberry Pi transmits the data it has received. The communication between each Raspberry Pi and the server is managed by Flask.

Using this hardware setup, a room is configured with objects such as tables or other furniture, along with human agents, to simulate various scenarios. One or more sensors are strategically placed to capture thermal data, enabling accurate evaluation and comparison with the simulated results.

### Metrics

To validate the correspondence between the temperature images produced by the simulator and those produced by the real sensor, we used three metrics: correlation between images, mutual information, and the evaluation graph.

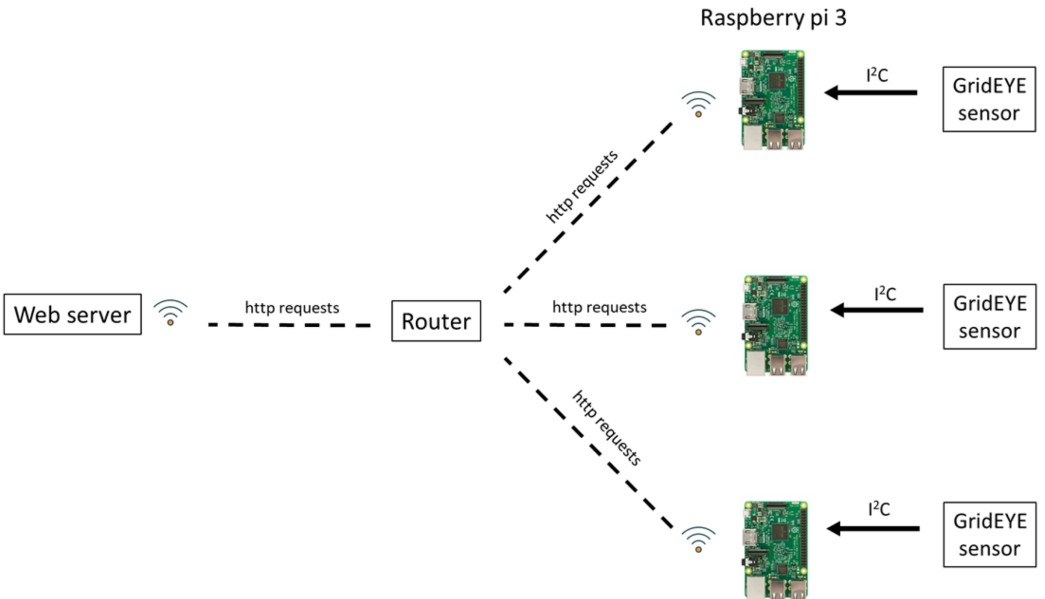

**Figure 8** System overview (images CC0 1.0 and PD from https://commons.wikimedia.org/).

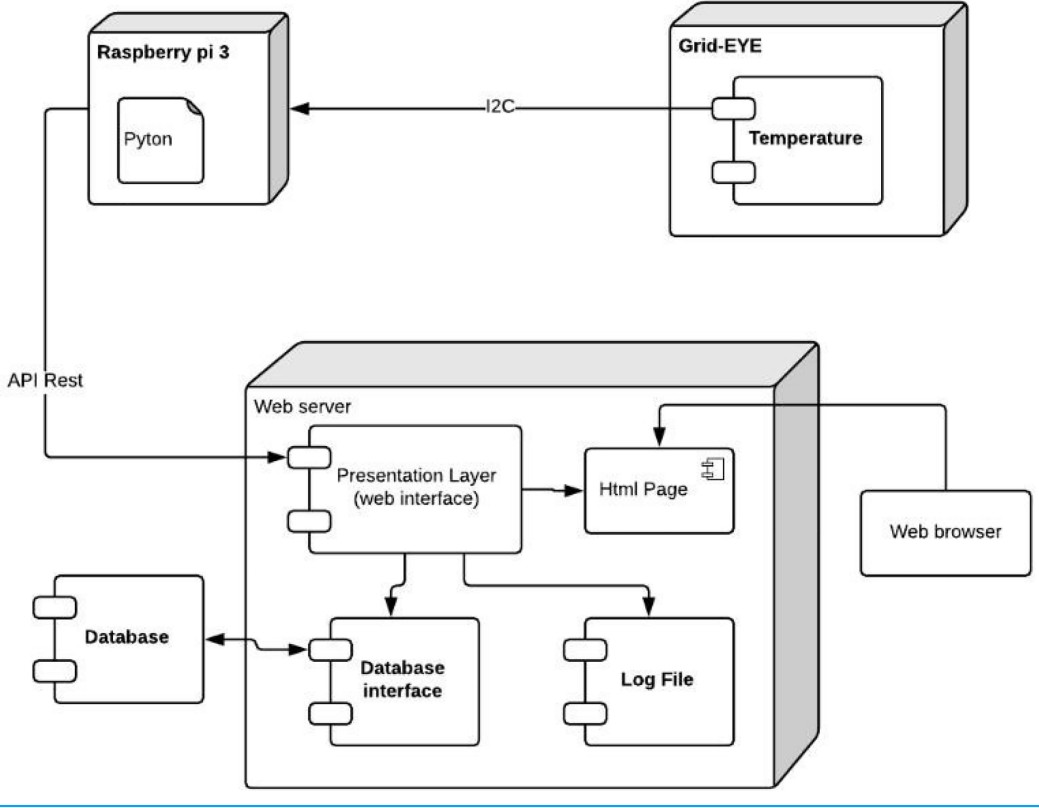

**Figure 9** Component diagram of the system.

### Correlation

In statistics, correlation measures how two quantitative variables are related. It indicates the extent to which a change in one variable is associated with a change in another variable. This relationship can be positive (both variables increase or decrease together), negative (one variable increases while the other decreases), or zero (no apparent relationship).

The most common measure of correlation is the Pearson correlation coefficient ($r$) represented by Eq. (1), which ranges from −1 (perfect negative correlation) to +1 (perfect positive correlation), with 0 indicating no correlation.

$$-1 \leq r = \mathrm{corr}(X, Y) = \frac{\sum (x_i - \mu_X)(y_i - \mu_Y)}{\sqrt{\sum (x_i - \mu_X)^2 \sum (y_i - \mu_Y)^2}} \leq +1. \tag{1}$$

Given two images X and Y, $x_i$ and $y_i$ represent the $i^{th}$ pixel values in the images. $\mu_X$ and $\mu_Y$ denote the average pixel values of the images X and Y, respectively.

The Pearson correlation coefficient can be used to analyze the correlation between two images by comparing the intensity values of corresponding pixels in the two images. Specifically, the closer r is to 1, the more similar (correlated) the images are. Instead, the closer r is to 0 (or less than zero), the weaker the correlation, and the images are dissimilar.

### Entropy and mutual information

In information theory, the entropy of a message represents the average amount of information contained in the message when emitted (*Shannon, 1948*). It measures the uncertainty or unpredictability of a message: If a message is highly predictable, meaning the same content is repeated, the entropy is low because little new information is provided. Conversely, if each message is highly varied and unpredictable, the entropy is high, reflecting the novelty and amount of information conveyed.

For a discrete random variable X, the entropy is defined as:

$$H(X) = - \sum p_X(x_i) \cdot \log_2 p_X(x_i) \tag{2}$$

where $p_X(x_i)$ is the probability that the expected value is $x_i$.

Figure 10 illustrates various entropy concepts. The area shared by both circles represents the *joint entropy* $H(X, Y)$. The circle on the left corresponds to the individual entropy $H(X)$, and the circle on the right corresponds to $H(Y)$. The overlapping portion represents *mutual information* $I(X;Y)$.

The joint entropy of two variables X and Y, represented by $H(X, Y)$, is given by:

$$H(X, Y) = - \sum p_{XY}(x_i, y_i) \cdot \log_2 p_{XY}(x_i, y_i) \tag{3}$$

where $p_{XY}(x_i, y_i)$ is the joint probability of X taking the value $x_i$ and Y taking the value $y_i$. If X and Y are independent, their joint probability $p_{XY}(x_i, y_i)$ factors into the product of their marginal probabilities:

$$p_{XY}(x_i, y_i) = p_X(x_i) \cdot p_Y(y_i). \tag{4}$$

However, when X and Y are dependent, their joint entropy is less than the sum of their entropies. For example, when comparing two images X (the real one) and Y (the simulated one), the intensity of corresponding pixels in one image provides information about the

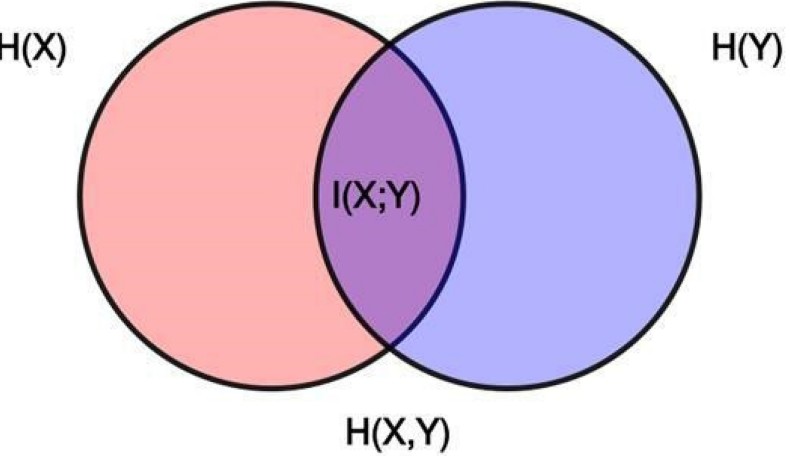

**Figure 10 Graphic visualization of the relationship between the entropy of two statistical variables and the mutual information $I(X, Y)$.**

intensity in the other image. It follows that the following relationship holds:

$$H(X, Y) < H(X) + H(Y). \tag{5}$$

Given two variables $X$ and $Y$, mutual information ($I(X; Y)$) quantifies the amount of information that one variable provides about the other. It provides a measure of their correlation and is directly defined in terms of their entropies: the higher the mutual information, the more the variables share information. $I(X; Y)$ measures the difference:

$$I(X, Y) = H(X) + H(Y) - H(X, Y). \tag{6}$$

Mutual information $I(X; Y)$ reaches its maximum value when $X$ and $Y$ are perfectly related, *i.e.*, when I(X;Y) = H(X) = H(Y). I(X, Y) is minimum (*i.e.*, equal to 0) when the distributions of pixel intensities in the two images are independent of each other. This means that the pixel intensities in one image do not provide any useful information about the intensities of pixels in the other image, indicating a complete dissociation between simulations and real ones.

### Evaluation graph and image similarity

The evaluation graph provides a visual comparison of two images (real and simulated) by illustrating the differences at the pixel level. This graph helps to assess how closely the two images represent the same scenario based on temperature variations.

The value on the x-axis represents a temperature variation (in degrees), while the y-axis indicates the percentage of pixels where the difference in temperature between the real and simulated images is below the specified value on the x-axis.

The process for analyzing the differences is the following. First, a matrix is created that shows the absolute differences in temperature values for each corresponding pixel in the simulated and real images and a temperature threshold is set as explained below. If the value of a pixel in the difference matrix exceeds this threshold, its contribution is zero; otherwise, its contribution is one. The sum of these contributions, normalized by the total

number of pixels, provides an estimate of the likelihood that the two images represent the same scenario.

Using the characteristic sensor error (reported in the datasheet), the evaluation threshold used is $2 * \sigma$ (a restrictive condition where 95% of the values fall in this range). The percentage of pixels in the difference matrix below this threshold is the metric used to assess the similarity between the two images.

For example, in the case of the Grid-EYE, the error reported in the datasheet is ±2.5 °C. Modeling the error as a normal distribution with a mean of 0, 99.7% $(3 * \sigma)$ falls within this range, *i.e.*, $3 * \sigma = 2.5$ °C, from which $\sigma \approx 0.83$. The higher the percentage, the more similar the images are. For example, if the evaluation finds that 90% of the pixel differences fall below the threshold, this means that only 6 (of the $8 \times 8$) pixels differ between the simulated and real images. This low number suggests a high degree of similarity, confirming that the simulation closely resembles reality.

## EXPERIMENTS AND RESULTS

### Experiment setup

For the collection of real-world data, we used the Grid-EYE sensor AMG8833 characterized by the following parameters: $8 \times 8$, FOV 60°, temperature accuracy ±2.5 °C, and human detection distance 5 m (as from sensor datasheet). The real-world experiments were done in a single room with dimensions $3,200 \times 4,000 \times 2,700$ mm³. The Grid-EYE sensor was positioned at $x = 1,250$ mm, $y = 1,400$ mm, $z = 2,700$ with an elevation angle of $= 0°$ and azimuth angle of $= 90°$. The room contained the following elements:

- A large four-station table approximated with a parallelepiped $x = 3,200$ mm, $y = 1,600$ mm, $z = 750$ mm placed in the centre and with room temperature.
- A fan-coil ($x = 240$ mm, $y = 1,200$ mm, $z = 600$ mm) positioned at $x = 120$ mm, $y = 900$ mm, $z = 0$ mm. During the experiments, the fan-coil's temperature was varied to simulate different thermal conditions.
- An Agent 1,750 mm tall, 500 mm wide (measured across the shoulders), and 200 mm deep (from front to back).

In the real-world environment, thermal images were captured directly from the physical sensor. These images inherently included the effects of physical sensor characteristics and noise.

For the simulation, the environment and the sensor were modelled to replicate the same setup and behaviour as described above. Noise was modeled as a normal distribution with zero mean and a standard deviation of $\sigma = 0.2$ °C (the 99.7% confidence interval is ±0.6 °C, which corresponds to $3\ \sigma$).

We conducted several experiments in real-world settings and replicated them using the simulator, considering different scenarios.

For each experiment, we summarize the changing parameters (*e.g.*, the position of the agent and the temperatures of the various components) in a table. The table also reports

the results in terms of correlation (C), entropy and mutual information (EMI), and image similarity (IS) values between the real and the simulated images.

For static experiments, the results are calculated as the average of the metric values across 10 real and 10 simulated images. Due to the stationary condition, each real frame is compared with all 10 simulated frames, resulting in 100 metric values.

For dynamic experiments, only corresponding frames can be compared. The simulator calculates a trajectory based on the points of the polyline provided in a CSV file. The avatar follows this trajectory at a constant speed (a simulation parameter expressed in mm/sec). Based on the sampling frequency (a simulation parameter in Hz), the simulator identifies the sampling points along the trajectory. Using this information, the points and the corresponding times when the user must be at those points are determined. These two sets of images—real and simulated—are then used for comparison. It is worth noting that the duration of the simulation, under the same scenario, is determined by how long the avatar takes to complete the given path.

## Experiment 1—static, centered

This experiment was conducted in a static condition, with a standing person positioned below the Grid-EYE, aligned with the sensor axes. The top part in Table 1 reports all the setup parameters.

### Discussion

With a difference of 4.5 °C between the room environment and the person, and a temperature accuracy of ±2.5 °C, this experiment demonstrates that the simulation correctly replicates real sensors when a standing person is positioned in the middle of the field of view of Grid-EYE. The good correlation, high mutual information, and successful pixel-by-pixel comparisons reported in the second part of Table 1 validate the simulator's accuracy in reproducing real data under these specific conditions. The simulated image has higher entropy compared to the real one (5.2 *vs* 3.5), indicating greater pixel variability. This indicates that the noise model in the simulation introduces more noise than observed in the real data. The noise contributing to the increase in entropy in the simulated images is a controllable parameter within the simulator. This allows us to adjust it according to the desired fidelity level relative to real-world conditions. In practice, noise can be reduced to obtain more uniform thermal simulations consistent with real data by tuning the parameter of the standard deviation of the noise. However, experimental observations show that even with a high level of artificially introduced noise, the real and simulated activities remain clearly distinguishable. This suggests that although noise affects certain statistical metrics such as entropy, it does not substantially compromise the system's ability to discriminate between different activities, at least under the experimental conditions tested so far.

From the perspective of image entropy analysis, mutual information indicates a strong correspondence between the image produced by the simulator and the real one. As indicated by the numerical values, the mutual information is high even though one image has higher entropy than the other. This means that most of the additional variability

**Table 1 Experiment 1 set-up and results.**

| Experiment n.1 | | Standing person, centered below the Grid-EYE | |
|---|---|---|---|
| **Set-up** | **Values** | **Notes** | |
| Agent position (x, y, z) | (1,200, 1,350, 0) | | |
| Agent body temperature | 28 °C | Average person temperature with clothing | |
| Ambient temperature | 23.5 °C | Average ambient temperature | |
| Fan-coil temperature | 28 °C | | |
| **Results** | **Average values** | **Notes** | |
| C | 0.79 | | |
| EMI | 3.5; 5.2—5.9—2.8 | Entropy real images; entropy simulated images—Joint entropy—Mutual information | |
| IS | 95% | Pixel-wise differs less than $2 * \sigma = 1.66$ °C | |

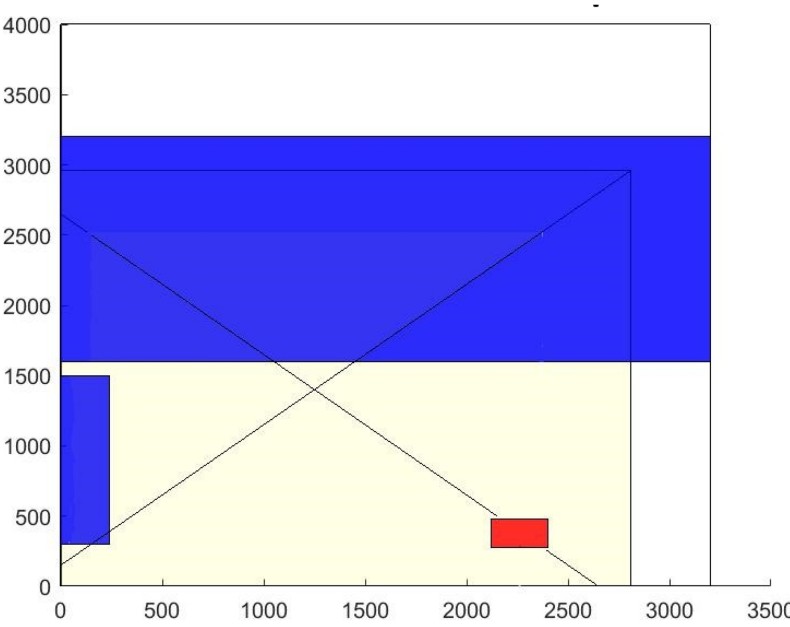

**Figure 11 Experiment 2: top view of the environment with the position of the person (small red rectangle on the bottom-right), other objects (blue, in the centre and left-hand side) and the sensor (yellow rectangle with its two diagonals).**

(introduced noise) has not significantly altered the shared information between the images. In other words, the images essentially contain the same relevant information.

## Experiment 2—static, corner

This experiment was conducted in a static condition: a standing person below Grid-EYE, in the corner, as shown in Fig. 11. Temperature parameters are reported in Table 2.

### Discussion

The results of this experiment, summarized in the second part of Table 2, present a strong correlation (0.71) and 90% image similarity indicating a high level of correspondence

**Table 2 Experiment 2 set-up and results.**

| Experiment n.2 | Standing person below Grid-EYE in the middle | |
|---|---|---|
| **Set-up** | **Values** | **Notes** |
| Agent position (x, y, z) | (2,300, 450, 0) | In a corner of the sensor view (Figure ref{corner}). |
| Agent body temperature | 26 °C | Average person temperature with clothing |
| Ambient temperature | 20 °C | Average ambient temperature |
| Fan-coil temperature | 27 °C | |
| **Results** | **Average values** | **Notes** |
| C | 0.71 | |
| EMI | 3.3; 4.5—5.9—1.9 | Entropy real images; entropy simulated images—Joint entropy—Mutual information |
| IS | 90% | Differs less of 2 * $\sigma$ = 1.66 °C |

**Table 3 Experiment 3 set-up and results (average).**

| Experiment n.3 | Person walking under Grid-EYE | |
|---|---|---|
| **Set-up** | **Values** | **Notes** |
| Human position (x, y, z) | From (300, 1,340, 0) to (2,800, 1,340, 0) | Figure 12 shows the trajectory with a red dotted line. |
| Human body temperature | 27 °C | Average person temperature with clothing |
| Ambient temperature | 23.5 °C | Average ambient temperature |
| Fan-coil temperature | 27.5 °C | |
| **Results** | **Values** | **Notes** |
| C | 0.58 | |
| EMI | 3.2; 4.7—5.9—2.0 | Entropy real images; entropy simulated images—Joint entropy—Mutual information |
| IS | 94% | Differs less of 2 * $\sigma$ = 1.66 °C |

between the simulated and real data. As reflected in the numerical values, the entropy of the simulated image is higher than that of the real image, supporting the observations from the discussion of Experiment 1 regarding the noise model. Despite this, the mutual information remains high, indicating that the images contain the same relevant information.

## Experiment 3—dynamic, sensor in the centre

This experiment was conducted under dynamic conditions: a person walking under the sensor, with the setup described in Table 3. The simulated scenario is depicted in Fig. 12. It is worth noting that in this experiment scenario, the impossibility of having perfectly synchronized frames naturally results in a higher error margin compared to the static scenario, where the positions of both the agent and the person can be precisely controlled and replicated. This increased error stems from the combined effect of sensor noise and the temporal misalignment between real and simulated frames. In dynamic scenes, even small differences in timing can lead to significant discrepancies in pixel-level comparisons, as the same physical action may appear slightly shifted or distorted across frames.

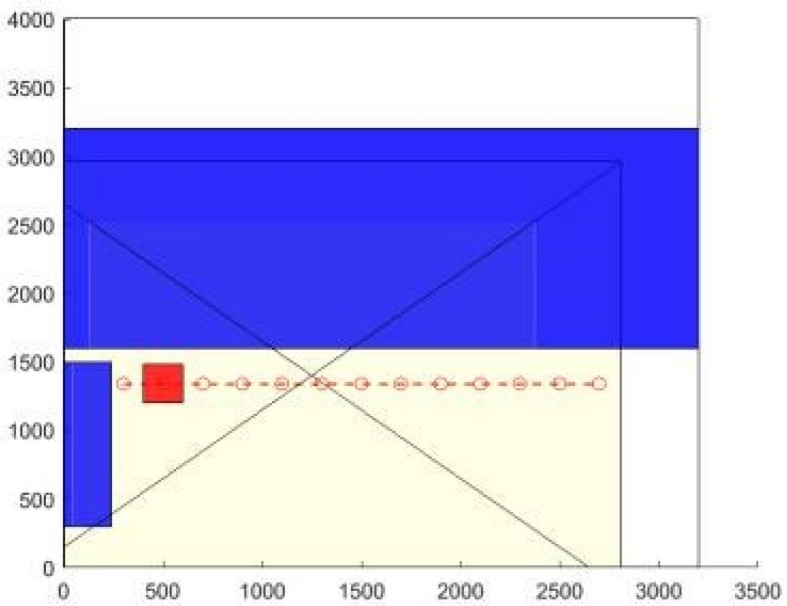

**Figure 12 Experiment 3: agent in movement (red-dotted line).** Objects and sensors are like in Experiment 2.

However, the primary goal of this experiment is not to achieve frame-perfect synchronization, but rather to demonstrate that the simulation produces behaviorally and spatially consistent outputs when compared to real-world observations.

Despite the lack of exact alignment, the real and simulated sequences display coherent motion patterns and plausible interactions, confirming that the simulator reliably captures the qualitative dynamics of the scenario. This supports its use as a valid approximation tool in downstream tasks such as activity classification, motion analysis, or scenario prototyping.

Figure 13 displays some significant frames from both the simulator and real sensor data, synchronized frame-by-frame.

### Discussion

The metric values in the second part of Table 3 validate the simulation's similarity to the real environment, even for dynamic scenarios. A limitation of the results is that the alignment of the frames was corrected manually, eliminating some frames that showed visible misalignment. This misalignment is due to the fact that the speed of the agent is uniform while the speed of the person is not. However, it is interesting to note that these imperfections are mitigated by the low resolution of the sensor.

### Experiment 4—dynamic, sensor angled and positioned left

This experiment was carried out under dynamic conditions, with a person walking in the room. As in the previous case, the sensor is ceiling-mounted but is now positioned to the left side of the room and inclined by 30°. Figure 14 shows the configuration of the

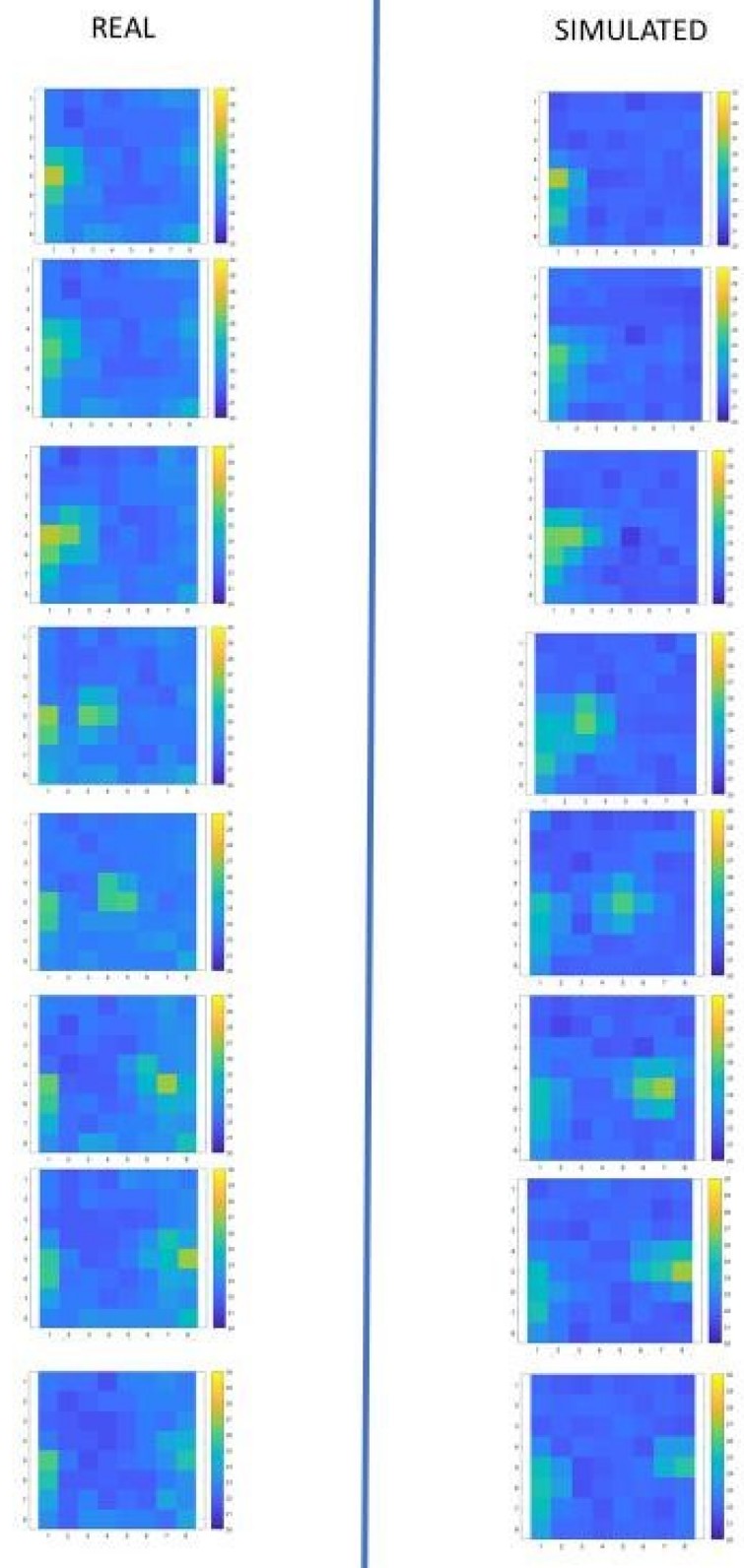

**Figure 13 Experiment 3: a subset of frames showing a person in different positions.**

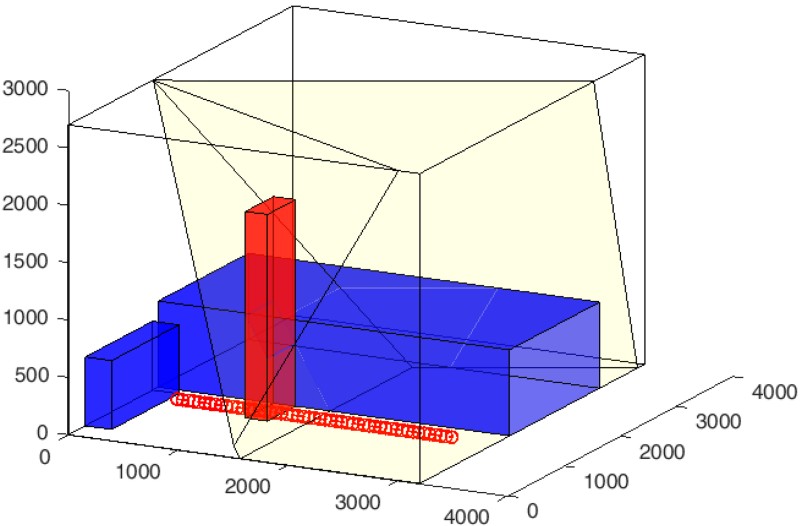

**Figure 14 Experiment 4: configuration with the sensor positioned in the left part of the room and inclined by 30°.**

**Table 4 Experiment 4 set-up and results (average).**

| Experiment n.4 | Standing person below Grid-EYE in the middle | |
|---|---|---|
| **Set-up** | **Values** | **Notes** |
| Human position (x, y, z) | From (300, 1,340, 0) to (2,800, 1,340, 0) | Figure 14 shows the trajectory with a red dotted line. |
| Human body temperature | 27 °C | Average person temperature with clothing |
| Ambient temperature | 21.0 °C | Average ambient temperature |
| Fan-coil temperature | 28.0 °C | |
| **Results** | **Values** | **Notes** |
| C | 0.49 | |
| EMI | 2.9; 4.6—5.9—1.6 | Entropy real images; entropy simulated images—Joint Entropy—Mutual information |
| IS | 92% | Differs less of $2 * \sigma = 1.66$ °C |

experiment and the person/agent's trajectory on the floor of the room. Details about the setup are reported in Table 4.

### Discussion

In this comparative analysis, the metric values summarized in Table 4 indicate that the simulation and real-world scenarios are compatible. As observed in Experiment 3, a limitation in the results verification process arises from the misalignment between frames, which was conducted manually. Again, this misalignment is attributed to the fact that the agent's speed is uniform, whereas the person's speed is not.

### Experiment 5—sensor with 80 × 60 pixels

The simulation environment is designed to analyze the effect and outcome of sensor placement with medium-low resolution and models only parallelepipeds. However, with

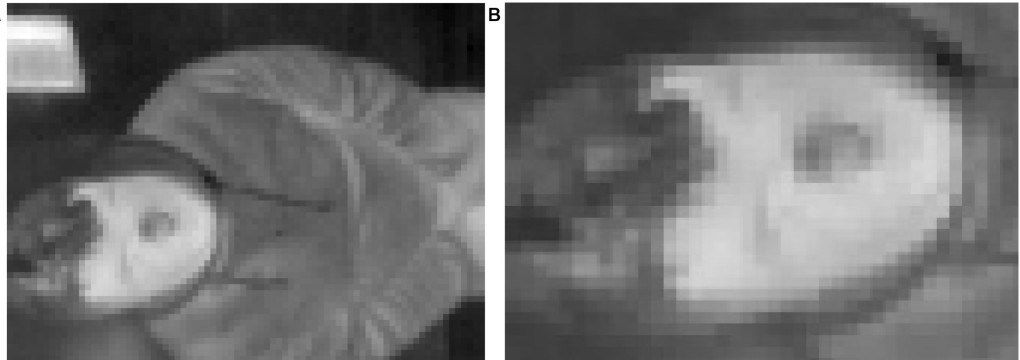

**Figure 15 Experiment 5: (A) thermal image of a person obtained from the LEPTON camera positioned on the laboratory ceiling at approximately 2.4 m. (B) Crop of the image on the face.**

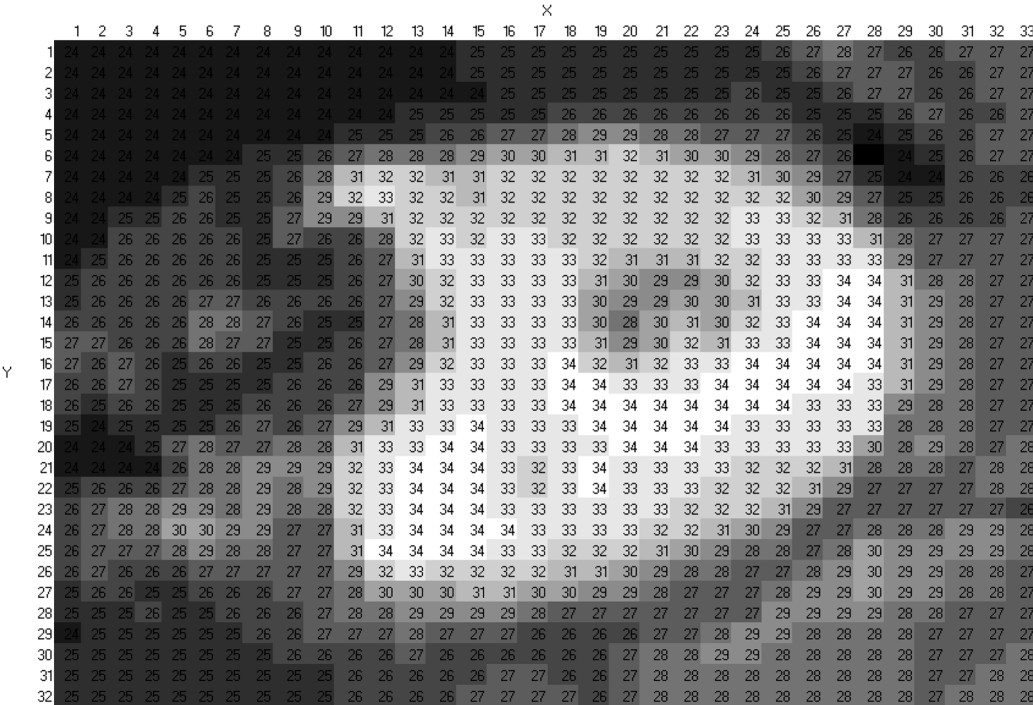

**Figure 16 Experiment 5: cropped image (32 × 33 pixels).** Each pixel is associated with an estimated temperature.

this experiment, we tested its configurability by accurately replicating fine-grained temperature gradients of a more complex object and a sensor with medium resolution.

In particular, we modelled a *FLIR LEPTON* camera with an 80 × 60 sensor, characterized by a standard deviation of 1.6 °C (high gain: ±5 °C at 25 °C), and a 50° × 50° FOV (*FLIR, 2025*). The modelled object was a face viewed from above. The face model was created starting from a portion of a thermal image captured with a real *FLIR LEPTON* camera (Fig. 15A), resized to approximately 32 × 33 pixels (Fig. 15B). Based on the color of

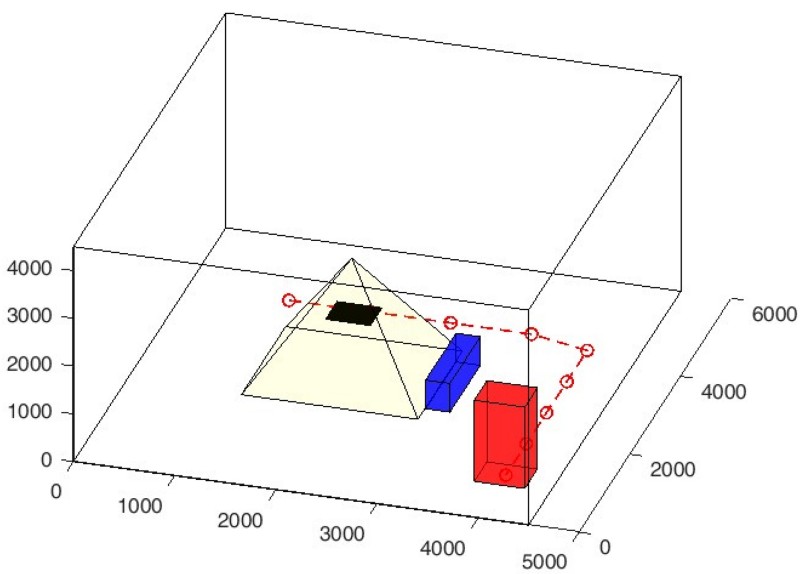

**Figure 17 Experiment 5: simulated environment with a 32 × 33 array of objects (9 mm × 9 mm × 10 mm each), forming a flat rectangular surface positioned 1,200 mm below the sensor.**

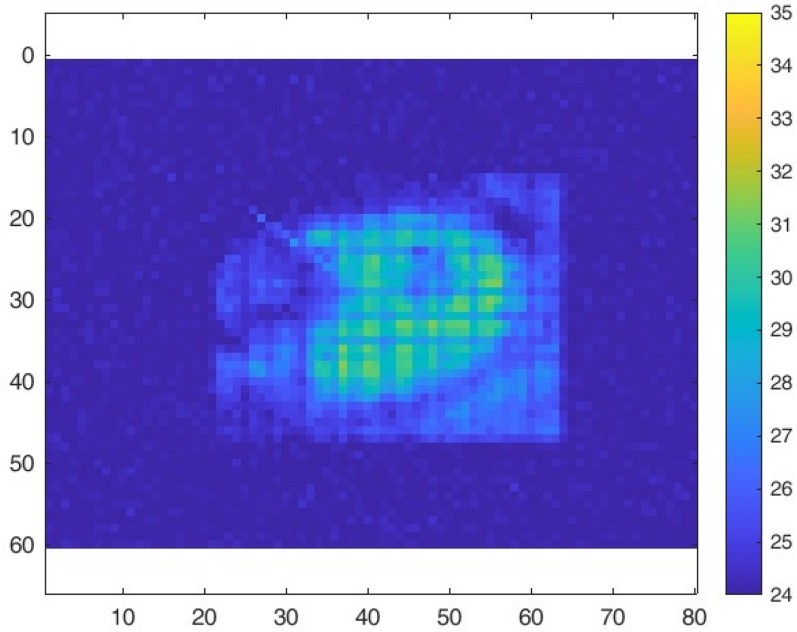

**Figure 18 Experiment 5: simulation result.** The face modelled with 32 × 33 objects is positioned at the centre of the sensor's field of view, with a resolution of 80 × 60.

the original image, a temperature was associated with each pixel, as shown in Fig. 16. Specifically, darker values were assigned a temperature of 23 °C, while lighter values were assigned 34 °C. In the simulation, each pixel is treated as a separate object,

**Table 5 Summary of the five experiments.**

| Experiment | Type | Description | Objective | Outcome |
|---|---|---|---|---|
| 1 | Static | Person standing in the center of the room, directly below the sensor. | Validate simulator's ability to replicate ideal and symmetric conditions. | Excellent match (C = 0.79, MI = 2.8, IS = 95%). Accurate sensor view reproduction under full visibility. |
| 2 | Static | Person placed in a corner of the sensor's field of view. | Test simulator under peripheral and asymmetric conditions. | Good match (C = 0.71, MI = 1.9, IS = 90%). Reliable in peripheral vision conditions. |
| 3 | Dynamic | Person walking in a straight line directly under the sensor. | Validate simulator in dynamic scenarios with continuous movement. | Acceptable match (C = 0.58, MI = 2.0, IS = 94%). Effective handling of temporal misalignment. |
| 4 | Dynamic | Person in motion with sensor placed laterally and tilted at 30°. | Test simulator with complex geometries and oriented sensor positions. | Acceptable match (C = 0.48, MI = 2.9, IS = 92%). Validates model flexibility and geometrical adaptability. |
| 5 | Static | Simulation using a higher resolution sensor (80 × 60) to test scalability. | Verify adaptability to different sensor types. | Simulator generates thermal images demonstrating scalability to varying resolutions. |

positioned in close proximity to others to reconstruct the complete face model with spatial continuity. The assembled face model (which becomes a flat complex object) is then placed in front of a virtual camera with resolution 80 × 60, as shown in Fig. 17. Figure 18 shows the thermal image generated by the simulator, demonstrating its ability to replicate also more complex real-world conditions and simulate cameras with varying resolutions.

## Summary of the experiments

The simulator was validated through five experiments (summarised in Table 5) comparing real sensor data with simulations in a digital environment. Among these, two experiments were conducted in static scenarios, two involved dynamic motion, and one tested a thermal sensor with a different resolution. These experiments served two purposes: (i) validating the accuracy and realism of the simulated thermal images compared to real sensor data, and (ii) demonstrating the simulator's flexibility and applicability across various environmental and sensing configurations.

The results of the validation phase demonstrated that the simulator can reliably reproduce real thermal sensor data in both static and dynamic scenarios, across different sensor positions, configurations, and types. This confirms its potential as a valuable tool for the design and evaluation of systems in applications such as people detection, activity recognition, and the prototyping of smart environments.

## SCENARIO ANALYSIS

While the previous section focused on validating the simulator's accuracy through experiments comparing simulated and real data, this section illustrates two practical use cases of the simulator, with different sensor installation configurations—such as position, angle, and quantity: the first focuses on super-resolution using an array of sensors, the second addresses the detection of presence in and out of bed.



**Figure 19** **Examples of overlapping of two sensor grids.**

The two selected application scenarios—super-resolution and position detection in the bed area—were chosen to highlight different sensor placement strategies and their impact on system behaviour. In the super-resolution example, the sensors are positioned very close to each other, creating challenges in evaluating their individual contributions and how their combined fields of view affect the final result. This scenario is useful to demonstrate how the simulator handles fine-grained integration of overlapping data. Conversely, in the position detection in the bed area example, the sensors are placed at a distance and carefully angled to create overlapping views that compensate for blind spots. This highlights the simulator's ability to support heterogeneous and spatially distributed sensor setups, which are common in real-world smart environment deployments.

## Scenario A: super-resolution

In this scenario, we simulate the super-resolution technique. This technique enables the utilization of low-cost, low-resolution sensors to achieve the functionality of higher-resolution sensors. By strategically positioning multiple sensors in close proximity to each other, they can record overlapping data. This overlapping information can then be combined to create a virtual sensor that effectively mimics the capabilities of a higher-resolution device, as shown in Fig. 19.

Simulation facilitates experimentation, allowing users to adjust factors such as the number of sensors deployed and the distances between them. By testing these configurations, we can assess the effectiveness and improvements gained from utilizing multiple sensors in close coordination in detecting individuals' profiles.

For illustrative purposes, we use the $8 \times 8$ Grid-EYE sensor. This choice does not affect the general applicability of the approach.

One method of constructing a virtual sensor is to arrange an $n \times m$ grid of equally spaced sensors, as exemplified in Fig. 20.

As introduced in 'Related Work', temperature images are computed on the projection plane, which is orthogonal to the vertical axis, at operating distance $h$ from the sensor position (Fig. 21).

The distance between sensors $x_x$ and $x_y$ depends on the *operating distance* $h_x$ and $h_y$, the field of view of the sensor $FOV_x$ and $FOV_y$, the sensor resolution $R_x$ and $R_y$ and, the super-resolution $SR_x = R_x * k_x$ and $SR_y = R_y * k_y$. The sensor super-resolution (SR) is the

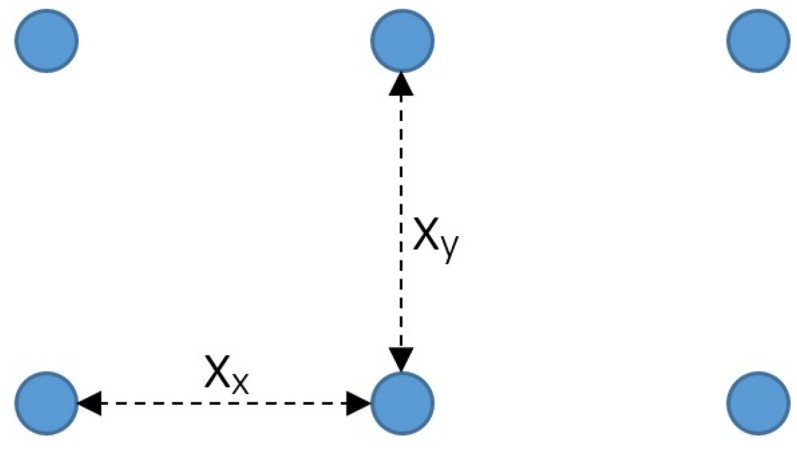

**Figure 20 Schematic representation of the positioning of a 3 × 3 sensor array.**

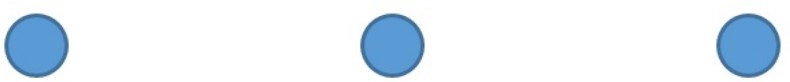
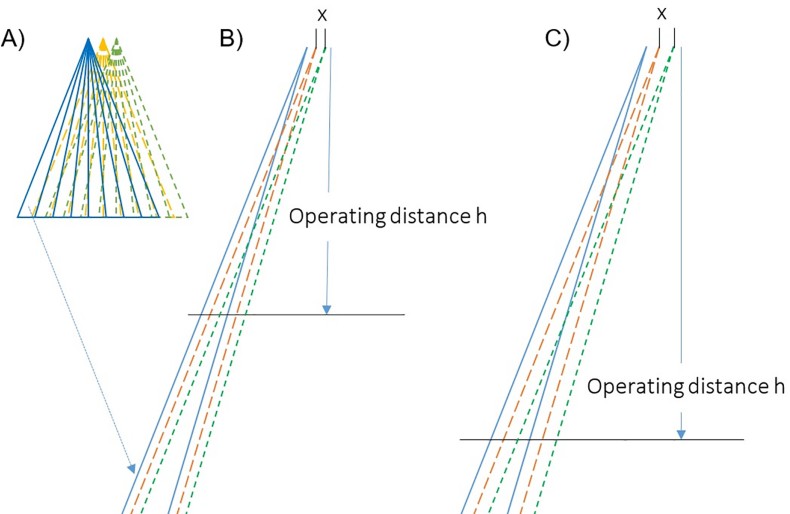

**Figure 21 Impact of different operating distances on sensor placement.** (A) shows three overlapping sensors; in (B) and (C) the positions of their leftmost slice are displayed for two different operating distances. Note that these operating distances depend on the sensor distance ($x$).

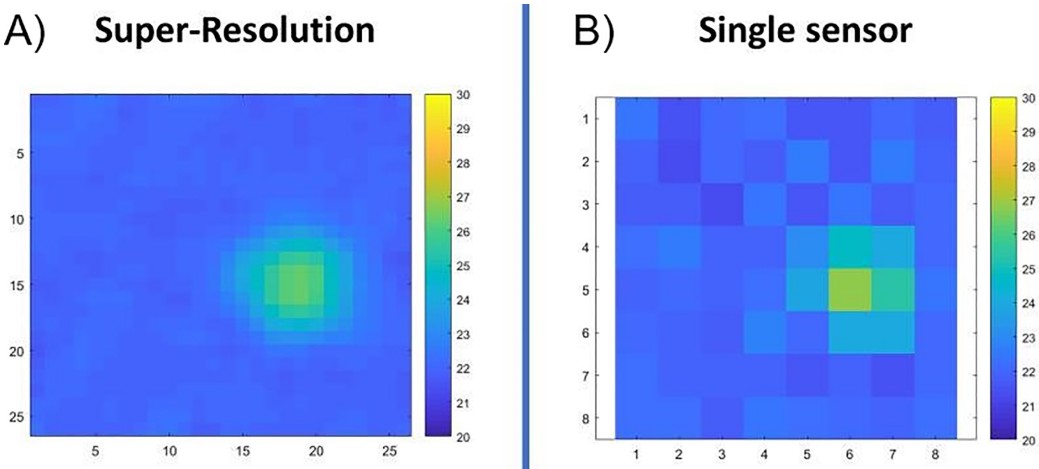

**Figure 22 Comparison of a high-resolution image with 3 × 3 sensors (A) with the image with a single sensor (B).**

sensor resolution $R$ multiplied by the desired number of pixels $k_x$ and $k_y$; the total number of sensors are $NumSensors = k_x * k_y$. It is important to emphasize that, unless specific requirements require otherwise, $h_x = h_y = h$.

Equation (7) determines the value of $x_-$ (for x o y axes).

$$x_- = \frac{2 * h_- * tan(\frac{FOV_-}{2})}{R_- * k_-} \tag{7}$$

It is important to emphasise that objects detected at distances different from $h$ will inevitably appear distorted in the perception of their size and/or shape.

Depending on specific requirements, having a different resolution per axis is possible, but the vision plane (determined by the operating distance) must remain the same. Another effect is the reduction of noise; specifically, the root mean square in every single pixel in the super-resolution decreases by a factor of $\sqrt{n}$, where $n$ is the number of overlaps. For the final resolution, the number of pixels is given by Eq. (8).

$$SR_x \times SR_y = ((R_x + 1) \cdot k_x - 1) \cdot ((R_y + 1) \cdot k_y - 1). \tag{8}$$

For example, using nine sensors (FOV: $60° \times 60°$, $R$: $8 \times 8$) arranged in a $3 \times 3$ grid with an operating distance of 2 m and a uniform multiplication factor of 3, the distance between the sensors is 9.6 cm, resulting in a super-resolution of $SR : 26 \times 26$. Similarly, using nine sensors (FOV: $60° \times 60°$, $R$: $8 \times 8$) arranged in a $3 \times 3$ grid with an operating distance of 2 m, a multiplication factor of 3 along the X-axis, and 2 along the Y-axis, the distance between the sensors is 9.6 cm along the X-axis and 14.4 cm along the Y-axis. The pixels are rectangular, and the resolution is $SR$: $26 \times 17$.

Figure 22 compares the results of the simulation of a $3 \times 3$ matrix with the simulation of a single sensor.

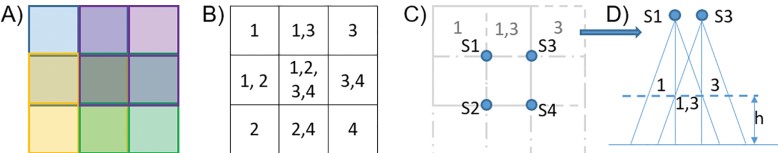

**Figure 23 Example of high-resolution image reconstruction.** From left to right: the final image (A), the real sensor pixels contributing to each virtual pixel (B); the four overlapping 2 × 2 grids (C); the overlap between sensors S1 and S3 (D).

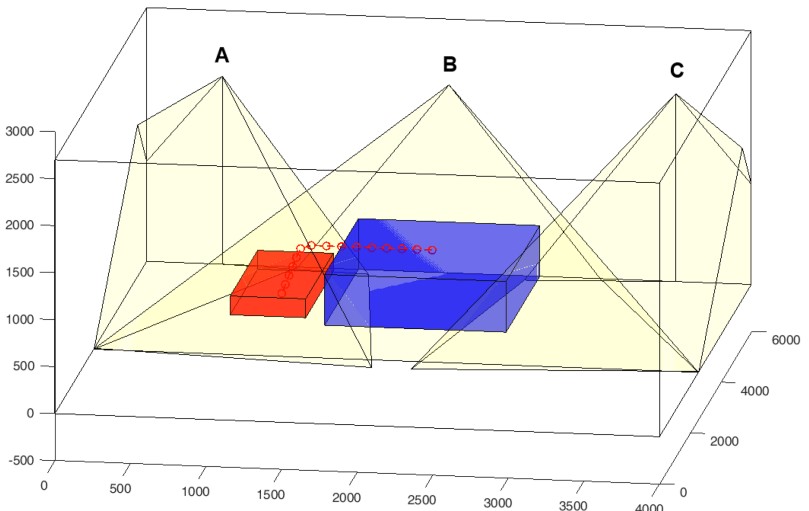

**Figure 24 Example of a person (red parallelepiped) falling from the edge of the bed (blue parallelepiped).** The red circles indicate the simulated trajectory where thermal images are captured.

The temperature for each virtual pixel is estimated by combining the temperature values detected by the overlapping real sensor pixels using methods such as averaging, interpolation, or selecting the maximum value.

For example, consider four sensors with a resolution of 2 × 2 used to generate a higher-resolution grid of 3 × 3 virtual pixels, as depicted in Fig. 23. This figure illustrates how the temperature values for the 3 × 3 virtual pixel grid are estimated using the real pixels from four sensors (S1 to S4). These sensors are placed at the operating distance $h$, such that the overlap between two adjacent sensors aligns with one real pixel. Each virtual pixel is calculated as the average of the temperature values from the corresponding overlapping real pixels.

## Scenario B: position detection in the bed area

In this scenario, the objective is to explore how to position and orient three low-resolution thermal sensors to effectively detect the position of a person in the bed area. This setup could be particularly useful for identifying critical situations, such as detecting if a person has fallen out of bed while sleeping.

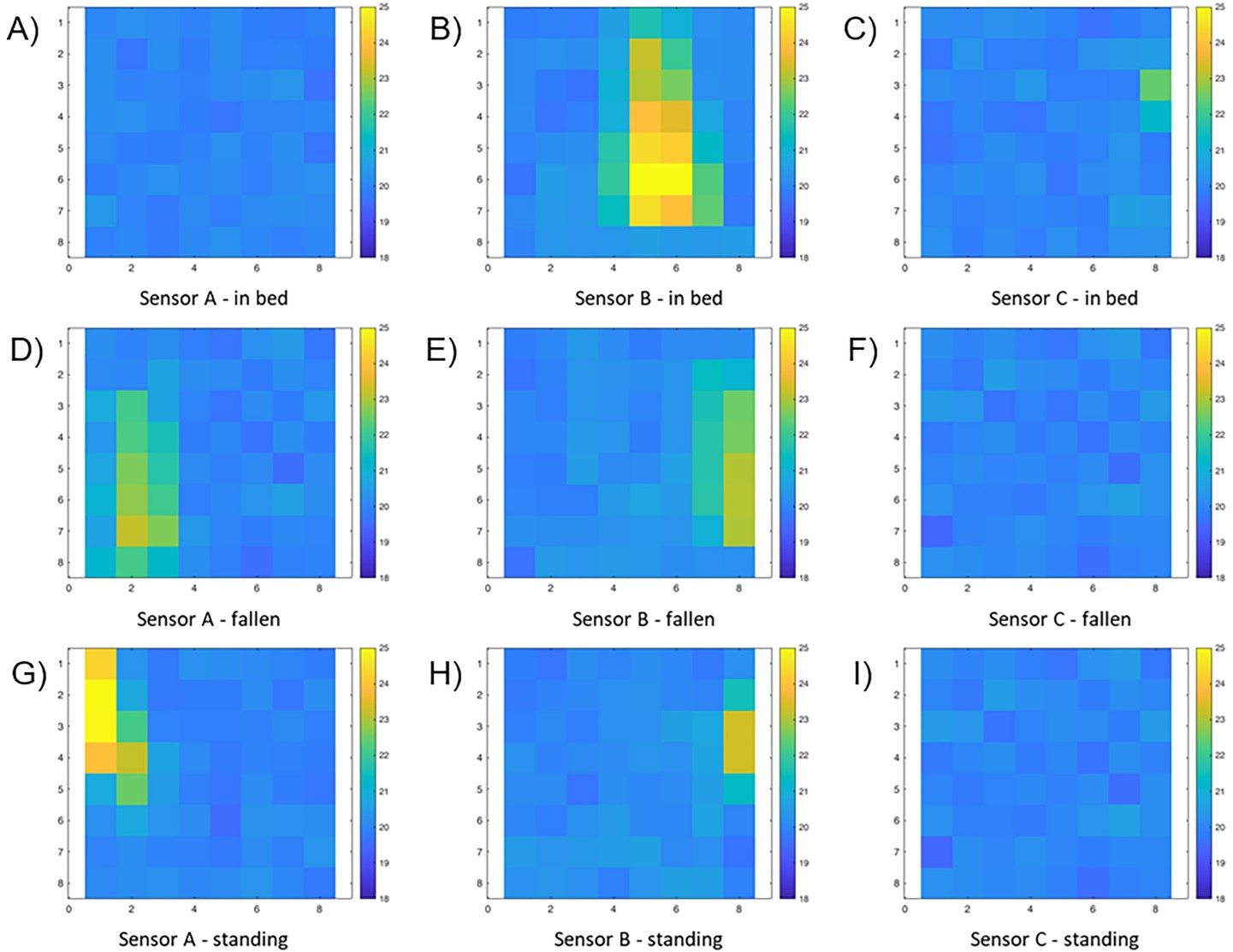

**Figure 25 Thermal images from the three sensors: before the fall (A–C), after the fall (D–F), and when standing at the edge of the bed (G–I).**

In an initial simulation, a single Grid-EYE sensor was positioned vertically above the bed. However, the results showed that detecting a fall based solely on temperature differences may not be reliable, because the change in height (around 0.60 m at most) when someone falls out of bed does not cause a significant enough temperature variation detected by the sensor. The small difference could be misinterpreted as part of the sensor's normal detection error, leading to missed detections.

An improved configuration to enhance the reliability of the monitoring system is shown in Fig. 24. In this setup, one sensor monitors the bed area (cone B), while the other two sensors are placed to monitor the bed's perimeter (cones A and C).

This arrangement of sensors enables the study of tracking effects in various situations: when a person goes to bed, tracking their movements during sleep, and detecting whether they get out of bed safely or fall.

In particular, sensors A and C are slightly tilted toward the bed to cover the blind spots of sensor B. This intentional overlap between the lateral sensors and the central (bed) sensor increases detection reliability by introducing redundancy and eliminating blind areas. Moreover, the overlaps could be exploited to enhance the level of detail in the detection area through super-resolution techniques. An example is shown in the following figures. Figure 24 also depicts an avatar transitioning from a lying position on the bed to a lying position near the bed. The first row in Fig. 25 displays the thermal images captured by the three sensors before the fall, while the second row shows the images after the fall. The images reflect the sensor's perspective, meaning the layout is "flipped": the right appears left, and the front appears at the top. Finally, for completeness, the third row shows the case in which, the avatar is standing at the edge of the bed. It can be observed that both the number of pixels involved and their intensity are different, allowing a distinction between "fallen at the edge of the bed" and "got out of bed." The simulation thus enables the analysis of different scenarios and allows for the preliminary evaluation of the effectiveness and limitations of the algorithms intended for use. Moreover, it also facilitates the generation of thermal image datasets for training machine learning algorithms.

## AREAS FOR FURTHER DEVELOPMENT

While the simulator offers a robust framework for environmental modelling, it is important to acknowledge its limitations. They do not compromise the primary purposes of the simulator but represent areas for the development of more complex scenarios that will be addressed in the next versions of the tool. They include:

- Temperature gradients: The model does not account for objects with varying temperatures, which may impact sensor accuracy during heating or cooling phases. However, realistic scenarios can still be represented by configuring objects with different static temperatures.
- Room shape: The environment is restricted to a parallelepiped shape (useful for the most common indoor standard scenarios), limiting spatial configurations.
- Single agent simulation: The current version allows for the simulation of only one moving agent at a time, restricting more complex interactions. This functionality is sufficient for many practical applications, such as monitoring individual subjects in domestic or security contexts. However, the ability to simulate interactions between multiple agents is a future development area for multi-user applications.
- Uniform movement: Agents move at a constant speed, with no acceleration or deceleration modelled. While this represents an acceptable approximation in many scenarios, the introduction of accelerations and decelerations could make the simulations more realistic.

- Simplified shapes: Both the environment and the agent, which are modelled as cuboids, which is sufficient for many applications. If detailed features are needed, multiple smaller cuboids with assigned temperatures.

- Collision detection: The simulator does not address potential collisions between objects or between objects and agents. However, the current configurations support accurate and realistic positioning of elements within the environment. In fact, with a single agent, it is possible to manage collisions effectively by manually configuring its trajectory and ensuring it avoids obstacles.

- Residual heat: The effects of residual heat from agents (for instance, after sitting) are not considered in the simulation. However, this does not affect most envisioned applications.

## CONCLUSIONS AND FUTURE WORKS

The use of low-resolution thermal sensors for indoor monitoring presents several challenges, including system configuration (such as number, placement, and angling of sensors), detection limits (temperature variations between the environment, objects, and monitored subjects), and data processing (including algorithms for analysis and classification). To effectively address these challenges and identify the most suitable solutions for specific environments, this work has introduced a configurable simulator, providing a platform for pre-testing before real-world implementation.

The simulator's configurability extends to various aspects, including environmental parameters, thermal properties of objects, and sensor settings, like resolution, field of view, and noise characteristics. The possibility of modelling a mobile agent with customizable trajectories, speeds, and sizes makes the simulator a versatile tool for a wide range of application scenarios.

Experimental validation demonstrated the simulator's reliability by comparing its outputs with real-world data under both static and dynamic conditions. The results demonstrated a strong similarity, confirming the simulator's reliability in replicating the behaviour of thermal sensors in real-world scenarios. Developed at the ATG Laboratory of Politecnico di Milano, the simulator has already been effectively applied in a home monitoring project as part of the BRIDGe initiative (with typical scenarios about elderly monitoring described in *Veronese et al. (2018*, *2016)*), which aims to promote autonomous and independent living.

The tool's adaptability allows it to address a variety of use cases, from optimizing sensor placement strategies to simulating complex monitoring environments, as illustrated by the usage scenarios presented in this work. By enabling pre-testing and refinement of configurations, the simulator can reduce implementation costs and improve the efficiency of real-world applications.

Finally, as part of future work, we plan to develop a graphical user interface (GUI) to enhance usability and user interaction.

The tool's code and examples are available at https://doi.org/10.5281/zenodo.14883451 (Version V1).

### Funding

This work was supported by MUSA: Multilayered Urban Sustainability Action, funded by Ministero dell'Università e della Ricerca (PNRR, Missione 4, componente 2, investimento 1.5). Project n. ECS 000037. No additional external funding was received for the remaining part. The funders had no role in study design, data collection and analysis, decision to publish, or preparation of the manuscript.

### Grant Disclosures

The following grant information was disclosed by the authors:
Multilayered Urban Sustainability Action.
Ministero dell'Università e della Ricerca: ECS 000037.

### Competing Interests

Sara Comai is an Academic Editor for PeerJ.

### Author Contributions

- Sara Comai conceived and designed the experiments, analyzed the data, prepared figures and/or tables, authored or reviewed drafts of the article, and approved the final draft.
- Andrea Masciadri conceived and designed the experiments, performed the experiments, performed the computation work, prepared figures and/or tables, authored or reviewed drafts of the article, and approved the final draft.
- Andrea Locati conceived and designed the experiments, performed the experiments, performed the computation work, prepared figures and/or tables, authored or reviewed drafts of the article, and approved the final draft.
- Alessandro Campi analyzed the data, prepared figures and/or tables, authored or reviewed drafts of the article, and approved the final draft.
- Fabio Salice conceived and designed the experiments, performed the experiments, analyzed the data, performed the computation work, prepared figures and/or tables, authored or reviewed drafts of the article, and approved the final draft.

### Data Availability

Code is available at Zenodo:
SALICE, F., Comai, S., & Masciadri, A. (2025). ATG Grid-eye simulator. Zenodo. https://doi.org/10.5281/zenodo.14883451.

### Supplemental Information

Supplemental information for this article can be found online at http://dx.doi.org/10.7717/peerj-cs.3124#supplemental-information.

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
