# Peer review of "A configurable simulator for low-resolution infrared thermal sensors: accuracy assessment and practical applications in indoor environments"

_PeerJ Computer Science, doi:10.7717/peerj-cs.3124_

## Round 0.1 · original submission · Major Revisions

**Language Note:** The review process has identified that the English language must be improved. PeerJ can provide language editing services - please contact us at [email protected] for pricing (be sure to provide your manuscript number and title). Alternatively, you should make your own arrangements to improve the language quality and provide details in your response letter. – PeerJ Staff

Reviewer 1 ·

Basic reporting

no comment

Experimental design

no comment

Validity of the findings

1. In Experiment 1, the entropy value of the simulated thermal image (5.2) was noticeably higher than that of the corresponding real image (3.5). This disparity implies that the simulation introduces additional variability or artificial noise that is not present in the sensor data. The elevated entropy level suggests that the simulation may not fully capture the thermal uniformity observed in real-world measurements under static and controlled conditions. Consequently, this limitation in accurately preserving thermal consistency could impact the reliability of the simulation when applied to subsequent tasks, such as human activity classification or thermal image-based detection. Please clarify this.

2. In Experiment 3, the results suggest an explicit limitation related to the absence of precise frame synchronization. Due to the experiment's inherently dynamic setting, achieving exact alignment between the simulated and real-world frames was not feasible. This lack of temporal synchronization leads to inconsistencies that may increase the error margin, complicating accurate frame-by-frame comparisons and undermining the evaluation of time-dependent elements such as motion patterns and trajectory analysis. Please clarify this.

Additional comments

1. The introduction provides a general overview of sensing technologies but does not identify the research gap or the precise aims of the present study. A well-defined problem statement would assist readers in understanding the study's context. Additionally, it omits comprehensive technical explanations and comparative evaluations of current sensors, resulting in uncertainty regarding the criteria for sensor selection or the metrics employed to assess their performance.

2. Within the related work section, it is recommended that the authors organize prior research into clearly defined subsections to enhance clarity and thematic coherence. Suggested thematic groupings include:
- Thermal Sensors for Activity Recognition and Occupancy Detection: This subsection should cover investigations that utilize thermal sensing technologies to identify and classify human activities. Illustrative examples of such studies include those conducted by Hand et al. (2022) and Puuruenn et al. (2021).
- Applications in Human Presence and Posture Detection: This category should encompass research that concentrates on detecting human presence and analyzing specific postural states or detailed body movements. Notable contributions in this area are found in the works of Trofimova et al. (2017), Mashiyama et al. (2015), and Shetty et al. (2017).
- Advancements in Thermal Sensing Algorithms and Optimization Methods: This section should compile research proposing algorithmic innovations to enhance sensor performance. Studies focusing on noise suppression, adaptive environmental modeling, and Kalman filter applications—such as those by Trofimova et al. (2017) and Shetty et al. (2017)—are especially relevant here.
- Identified Gaps and Challenges in Existing Approaches: Finally, a subsection should summarize the limitations highlighted in previous research, including issues like sensor positioning, spatial coverage, and calibration precision. It should also articulate the research gap addressed by the present work, namely, the lack of comprehensive strategies for optimizing sensor deployment in complex indoor environments.

·

Basic reporting

1. The proposed simulator's main objective is not clear enough. Authors are suggested to add the real problems that should be addressed. Further, some illustrations may be needed.
2. The authors are suggested to describe the simulator's user interface to better help readers understand.

Experimental design

1. The authors conducted five experiments, discussing and analyzing each result. However, the complete experiments are suggested to be summarized,
2. Since the experiments only used two thermal cameras, more justification should be made for handling other cameras, including the standard or high-resolution thermal camera.

Validity of the findings

The authors show two scenarios (Super-Resolution and Position Detection in the Bed Area) as examples for practical applications. In addition to the other examples, the reason for using these examples is suggested to be described.

Additional comments

The work is interesting and shows a promising application.

---

## Round 0.2 · accepted · Accept

Dear authors, we are pleased to verify that you meet the reviewer's valuable feedback to improve your research.

Thank you for considering PeerJ Computer Science and submitting your work.

Kind regards
PCoelho

Reviewer 1 ·

Basic reporting

In revising this manuscript, the author has made all the necessary corrections based on the provided suggestions. The work is complete and is deemed suitable for publication.

Experimental design

-

Validity of the findings

-

·

Basic reporting

The reviewer's comments have been addressed satisfactorily.

Experimental design

-

Validity of the findings

-